# Production and Fiber Characteristics of Colored Cotton Cultivares under Salt Stress and H_2_O_2_

**DOI:** 10.3390/plants12112090

**Published:** 2023-05-24

**Authors:** Luana Lucas de Sá Almeida Veloso, Carlos Alberto Vieira de Azevedo, Reginaldo Gomes Nobre, Geovani Soares de Lima, José Renato Cortez Bezerra, André Alisson Rodrigues da Silva, Reynaldo Teodoro de Fátima, Hans Raj Gheyi, Lauriane Almeida dos Anjos Soares, Pedro Dantas Fernandes, Vera Lúcia Antunes de Lima, Lúcia Helena Garófalo Chaves

**Affiliations:** 1Academic Unit of Agricultural Engineering, Federal University of Campina Grande, Campina Grande 58430-380, PB, Brazil; luana.lucas@estudante.ufcg.edu.br (L.L.d.S.A.V.); carlos.azevedo@profesor.ufcg.edu.br (C.A.V.d.A.); andre.alisson@estudante.ufcg.edu.br (A.A.R.d.S.); reynaldo.teodoro@estudante.ufcg.edu.br (R.T.d.F.); hans@pq.cnpq.br (H.R.G.); pedro.dantas@professor.ufcg.edu.br (P.D.F.); vera.lucia@professor.ufcg.edu.br (V.L.A.d.L.); lucia.garofalo@pesquisador.cnpq.br (L.H.G.C.); 2Department of Science and Technology, Federal Rural University of the Semi-Arid, Caraúbas 59780-000, RN, Brazil; reginaldo.nobre@ufersa.edu.br; 3Brazilian Agricultural Research Corporation, Campina Grande 58428-095, PB, Brazil; jose.cortez-bezerra@embrapa.br; 4Academic Unit of Agrarian Sciences, Federal University of Campina Grande, Pombal 58840-000, PB, Brazil; lauriane.almeida@professor.edu.br

**Keywords:** *Gossypium hirsutum* L., saline water, hydrogen peroxide

## Abstract

Salt stress reduces the yield and quality of colored fiber cotton production, but this problem can be mitigated by the foliar application of hydrogen peroxide in adequate concentrations. In this context, the objective of the present study was to evaluate the production and characteristics of fibers of naturally colored cotton cultivares under irrigation with low- and high-salinity water and leaf application of hydrogen peroxide. The experiment was carried out in a greenhouse under a randomized block design, arranged in 4 × 3 × 2 factorial scheme, corresponding to four concentrations of hydrogen peroxide (0, 25, 50, and 75 μM), three cultivares of colored fiber cotton (‘BRS Rubi’, ‘BRS Topázio’, and ‘BRS Verde’), and two electrical conductivities of water (0.8 and 5.3 dS m^−1^), with three replicates and one plant per plot. Irrigation with water of 0.8 dS m^−1^ associated with a foliar application of 75 μM of hydrogen peroxide favored the lint and seed weight, strength, micronaire index, and maturity of ‘BRS Topázio’. The ‘BRS Rubi’ cotton cultivar showed higher tolerance to salinity, followed by the ‘BRS Topázio’ and ‘BRS Verde’ cultivares regarding the yield of seed cotton weight, with reduction below 20% under water of 5.3 dS m^−1^.

## 1. Introduction

Cotton (*Gossypium hirsutum* L.) is an oilseed crop that constitutes the main source of natural fiber for the textile industries [1,2]. Thus, the yield and quality of cotton fibers are determining factors for its price in the market. Brazil is one of the largest cotton producers in the world, producing 2.26 million tons in the 2020/2021 season, in an area of 1.37 million hectares, reaching an average productivity of 1.65 t ha^−1^, with the northeast region responsible for 24.32% (0.574 million tons) of national production [3].

White fiber cotton requires dyeing, which raises production costs for the textile industry; in addition, it can cause damage to human health and the environment [4,5]. In this context, naturally colored fiber cotton has been cultivated and widely used by the textile industry, receiving attention from breeding programs, mainly in northeast Brazil, due to the technological differential of these fibers [6,7]. However, the quality of this fiber can be changed when the plant is subjected to abiotic stresses, such as salinity [8,9]. White fiber cotton has a salinity threshold in irrigation water of 5.1 dS m^−1^, but the genetic variability existing in the species, as well as temperature, time of exposure to stress, and management strategy, can influence the tolerance to salinity [10,11].

Irrigation with saline water can compromise molecular, biochemical and physiological processes, affecting plant growth, economic yield, and the quality of cotton fiber, due to the accumulation of salts, especially Cl^−^ and Na^+^, which cause a reduction in osmotic potential, ionic toxicity, nutritional imbalance, in addition to oxidative stress generated by the increased synthesis of reactive oxygen species (ROS) [10,12].

In this context, given the conditions of water scarcity, high temperature and high availability of saline water, which is characteristic in semi-arid regions, the use of genetic materials tolerant to salts and the application of hydrogen peroxide can help reduce the negative impacts of salts on plants and enable agricultural production. The exogenous application of hydrogen peroxide (H_2_O_2_) has been pointed out as a potential mitigator of the effects of salt stress on some crops such as soursop [13], passion fruit [14], pistachio [15], and cucumber [16].

Hydrogen peroxide is an ROS that, when applied at a low concentration in plants, before and after stress, can perform the function of signaling molecule and protection against biotic and abiotic stresses [17]. In addition, it can activate the defense system, contributing to a rapid adaptation of the plant to unfavorable conditions to its development [18,19]. In addition, H_2_O_2_ favors photosynthetic activity, improves the antioxidant metabolism and can also promote osmotic adjustment through the biosynthesis of cellular osmolytes such as proline, glycine-betaine, and soluble proteins [20,21].

This study is based on the hypothesis that the foliar application of hydrogen peroxide induces salt stress tolerance in fiber-colored cotton cultivares through the regulation of physiological and biochemical processes, which result in gains in production and fiber quality. Therefore, the objective of this study was to evaluate the production and quality of the fibers of naturally colored cotton under irrigation with water of low and high electrical conductivity and leaf applications of hydrogen peroxide.

## 2. Results and Discussion 

The multidimensional space of the original variables was reduced to two dimensions, represented by the first two principal components (PC1 and PC2) with eigenvalues greater than λ > 1.0 according to Kaiser [22]. The first two principal components made it possible to explain 81.9% of the variance contained in the original variables (Table 1); that is, PC1 and PC2 contributed with 48.41% and 33.49%, respectively. The third principal component, although with an eigenvalue higher than the unit, was not considered, since it did not add relevant information.

The variables with a correlation value greater than 0.6 (r > 0.6) were considered as relevant. Thus, in Table 2, the variables that had the highest discriminatory power in PC1 were seed cotton weight (SCW), lint cotton weight (LCW), percentage of fibers (%F), short fiber index (SFI), strength (gf/tex) (STR), micronaire index (MIC), maturity (MAT), and count strength product index (CSP). In PC2, the variables with the highest degree of correlation were the number of bolls (NB), boll weight (BW), fiber length (UHM), percentage of uniformity (UNF), and elongation at rupture (ELG).

It can also be noted in Table 2 that variables with the same sign act in a directly proportional manner, that is, when the value of one increases, the value of the other increases, or vice versa; and the variables with opposite signs act in an inversely proportional manner, so when the value of one increases, the value of the other decreases.

Based on the multivariate analysis of variance (Table 1), there were significant single effects of the factors hydrogen peroxide concentrations (H_2_O_2_), cotton cultivares (CG), and the electrical conductivity of water (ECw) (*p* ≤ 0.01). In addition, there was also a significant interaction between the factors of ECw and H_2_O_2_ concentration for the two PCs. However, the other interactions between the factors did not show a significant difference in any of the PCs.

The production components and fiber characteristics highlighted in the first two principal components (PC1 and PC2) are considered important for the selection of the H_2_O_2_ concentration and ECw, which can be used in the cultivation of naturally colored cotton in a protected environment in a semi-arid region of northeastern Brazil.

According to Figure 1A,B, the variables with a negative correlation were responsible for the peripheral discrimination of the treatment T142 to the right of PC1, and the variables with a positive correlation were responsible for the discrimination of treatment T241, to the left of PC1. In relation to PC2, the variables with a negative correlation were responsible for discriminating the treatment T121 at the lower part of PC2, and the variables with a positive correlation were responsible for discriminating the treatment T143 at the upper part of PC2.

In the principal component1, the treatment T142, which corresponds to the cultivation of ‘BRS Topázio’ irrigated with water of 0.8 dS m^−1^ and foliar applications of 75 μM of H_2_O_2_, was responsible for the highest values of LSW (57 g), LW (24 g), STR (35.85 gf/tex), MIC (4.65) and MAT (0.88) (Table 3). Conversely, the treatment T122 promoted the highest values of %F (44.90%) and CSP (3254.80); the highest value of short fiber index—SFI (12.95), was found in T141, which refers to the irrigation of ‘BRS Rubi’ cotton with water of 0.8 dS m^−1^ and applications of 75 μM of H_2_O_2_ (Table 3).

Comparing the results mentioned above with those found in the control treatment (T112), there were increments of 15.78% (LSW), 12.5% (LW), 0.31% (UNF), 2.11% (STR), 9.24% (MIC) and 1.13% (MAT). For the treatments T122 and T141, when compared with the respective controls (T112 and T111), there were increments of 2.56% (%F), 3.62% (CSP), and 9.18% (SFI) (Table 3).

In PC1 it can also be observed that the lowest values for LSW (29 g), LW (9 g), %F (25.64%), MIC (2.59), MAT (0.83) and STR (22.33) were observed in the treatment T123; and the lowest values for SFI (6.37) and CSP (1719.56) were observed in the treatments T142 and T141, respectively (Table 3).

Despite not having the best performance in relation to fiber quality, ‘BRS Topázio’ cotton plants irrigated with water of 5.3 dS m^−1^ and under applications of 25 μM showed satisfactory results when compared to the treatments that had the best values.

In analyzing the PC2, it was verified that the treatment T121 promoted the highest values of NB (14), BW (77.03 g), and ELG (8.33); the treatment T142, on the other hand, caused higher values of UHM (29.57) and UNF (86.75). However, the lowest values for NB (7) and BW (47.98 g) were verified in the treatment T143; on the other hand, the lowest value of ELG (4.90) was observed in the treatment T123; while the lowest values for UHM (20.73) and UNF (80.12) were found in plants under treatment T141.

In the PC2, it can also be observed that the treatments T231 and T222 showed adequate values of the variables belonging to this PC when compared to the treatments that had the highest values.

Foliar application with the highest concentration of H_2_O_2_ (75 μM) resulted in increased seed cotton weight (SCW), lint cotton weight (LCW), and resistance of the fiber to rupture (STR) in ‘BRS Topázio’ cotton when irrigated with low-salinity water (0.8 dS m^−1^). Generally, satisfactory results are not observed with the application of high concentrations of H_2_O_2_ in plants that are not under some type of stress [23,24], which was not found in the present study for the variables LSW, LW and STR. It is probable that this result is related to the time interval without H_2_O_2_ application to which the plant was subjected, considering that the last application was performed before the opening of the boll of the first plant; that is, a period of approximately 50 days without the application of H_2_O_2_. Thus, the hydrogen peroxide accumulated in plants that received 75 μM may have contributed to the increase in these variables, since H_2_O_2_ can accumulate in the plant and play a role in cell expansion, fiber development, and is also involved in the differentiation of the secondary wall of cotton fibers [25].

The foliar application of 75 μM of H_2_O_2_ increased the micronaire index (MIC) and maturity (MAT) of the fibers of ‘BRS Topázio’ cotton irrigated with water of 0.8 dS m^−1^ (Table 1). The micronaire index is an index of the thickness and maturity of fiber. Thus, MIC values lower than 3.5 are related to immature fibers, which are more likely to form neps in the yarn and in the finished fabric, as well as variations in dyeing [26]. However, micronaire index values above 5.0 are classified as thick and above the tolerable level by the market, depreciating the fiber value [27].

The maturity (MAT) of cotton fibers also increased in this treatment (T142); that is, the application of 75 μM of H_2_O_2_ increased the cell wall thickness of cotton fibers. Araújo [28] stated that the maturity characteristic (MAT) is influenced according to the deposition of cellulose layers in the fiber, and may vary according to climatic conditions, as well as pest attacks and early or late harvest. In the present study, there was an attack of pests (whitefly), but it did not compromise the maturity of the fibers. In addition, fibers with values above 0.83 are classified as mature according to the standardization of the textile industry, as they have better quality and are more suitable for processing [29].

In addition, Potikha et al. [30] reported that H_2_O_2_ can act as a sign of development in the differentiation of secondary walls in cotton fibers. These authors also pointed out in their study that the H_2_O_2_ production period coincided with the beginning of secondary wall deposition; the inhibition of H_2_O_2_ production or elimination of H_2_O_2_ available in the system prevented the process of wall differentiation; and the exogenous addition of H_2_O_2_ promotes secondary wall formation in young fibers.

Regarding the percentage of fibers (%F) and the count strength product index (CSP), it was found that the highest values were found in ‘BRS Topázio’ cotton plants irrigated with water of 0.8 dS m^−1^ and under H_2_O_2_ applications of 25 μM (T122) (Table 3). The percentage of fiber is an important characteristic linked to yield, given the preference of herbaceous cotton producers for cultivars with a percentage of lint/fiber greater than 40%, as the price of cotton fiber is higher than that of cotton seed [31].

In turn, the mean values of the CSP of the cultivares ranged from 1719.48 to 3254.80, being classified as very low to very high, respectively. The CSP index is related to the strength of the yarns, essentially depending on the individual tenacity of the fibers. Thus, the H_2_O_2_ application of 25 μM promoted high CSP values in ‘BRS Topázio’ cotton (T122), which may represent losses in fiber quality, as it exceeds the standard values accepted by the textile industry, which are within the range of 2000 to 2500 according to Bachelier and Gourlot [32].

Regarding the short fiber index—SFI, it was observed in the present study (Table 3) that the foliar application of 75 μM of H_2_O_2_ promoted a high SFI in ‘BRS Rubi’ cotton, classifying its fibers as being of low commercial value, since the textile industry accepts values lower than 10.0%. Thus, values above this percentage cause difficulties in the spinning process, as they lead to the production of yarns of low quality and with neps. A high percentage of short fibers result in yarns with irregular thicknesses, which may break in the thinnest and weakest portions during spinning and weaving [33].

Most treatments showed good results for SFI, with values below 10.0%, including treatments with an ECw of 5.3 dS m^−1^. Paiva et al. [34], studying the fiber quality of ‘BRS Verde’ cotton irrigated with waters of different salinity levels, found no negative effect of salinity on the quality of cotton fiber. Soares et al. [35] observed a lower short fiber index in the cotton cultivar ‘BRS Topázio’, regardless of salinity management strategies, corroborating the results of the present study, in which the lowest value was found in treatment T142 (Figure 1A,B).

For the principal component2, it was observed that treatment T121, which corresponds to irrigation with water of 0.8 dS m^−1^ and the application of 25 μM of H_2_O_2_ in ‘BRS Rubi’, favored the number of bolls (NB) and boll weight (BW). Adequate concentrations of H_2_O_2_ may favor production due to the role played by H_2_O_2_ in plant physiological processes such as senescence, photorespiration, photosynthesis, stomatal movement, cell cycle, development, and the expression of some genes in the cells [36]. Sarwar et al. [25] found that the application of H_2_O_2_ improved boll weight in plants subjected to stress due to the protection of physiological processes.

The elongation of the fiber (ELG) of ‘BRS Rubi’ cotton was also favored by the application of 25 μM of H_2_O_2_, being within the standard desired by the textile industry (greater than 7%). According to Xiao et al. [37], the H_2_O_2_ is a type of reactive oxygen species (ROS) that can promote the significant elongation of the cell fiber.

Foliar spray with 75 μM of H_2_O_2_ resulted in a high fiber length (UHM) and a high percentage of uniformity (UNF) in ‘BRS Topázio’ cotton, increasing the commercial quality of its fibers and ensuring lower losses in the spinning process [34]. According to Silva et al. [38], fiber length may vary according to the cultivar, which was observed in the present study, since the application of 75 μM of H_2_O_2_ promoted long fibers in the cultivares ‘BRS Topázio’ and ‘BRS Verde’ (from 27.94 to 32 mm); however, it reduced fiber length in the ‘BRS Rubi’ cultivar (less than 25 mm), depreciating its commercial classification [39]. However, if the processing causes fiber breakage, it will result in a decrease in fiber length and uniformity with an increase in the short fiber index [40].

The cotton cultivares ‘BRS Rubi’, ‘BRS Topázio’, and ‘BRS Verde’ were classified as tolerant to a salinity of 5.3 dS m^−1^, with reductions of 6.97%, 8.33% and 19.51%, respectively, in relation to the yield of seed cotton weight, as they showed a reduction below 20% [41].

However, given the fiber quality parameters evaluated in the present study, it should be pointed out that satisfactory characteristics of longer fibers, with higher resistance for the spinning and baling processes, with a lower percentage of short fibers and a greater uniformity of length [39], can be attributed to the plants of treatment T142.

## 3. Materials and Methods

### 3.1. Location of the Experiment

The experiment was conducted between November 2020 and March 2021 under the conditions of a protected environment (a greenhouse), at the Academic Unit of Agricultural Engineering (UAEA), Federal University of Campina Grande (UFCG), located in Campina Grande, Paraíba, Brazil, whose local geographic coordinates are 07°15′18′′ S, 35°52′28′′ W, with an average altitude of 550 m. 

During the experiment, the data of average temperature and average relative humidity were observed and shown in Figure 2. It is worth mentioning that cultivation in a protected environment offers a number of advantages for crops. These include, for example, less leaching of nutrients from the soil, the more efficient control of pests and diseases, protection against weather conditions such as rain, wind, hail, cold and high rates of radiation, and precocity, consequently generating an improvement in the quality of the products [42]. However, under the conditions of a protected environment, the temperature indexes are higher and the relative humidity is lower in relation to the external environment.

### 3.2. Treatments and Experimental Design

The design adopted was randomized blocks, in a 4 × 3 × 2 factorial scheme, corresponding to four concentrations of hydrogen peroxide—H_2_O_2_ (0; 25; 50 and 75 μM), three cultivares of colored fiber cotton—CG (‘BRS Rubi’, ‘BRS Topázio’, and ‘BRS Verde’), and two electrical conductivities of water—ECw (0.8 and 5.3 dS m^−1^), resulting in twenty-four treatments, with three replicates and one plant per plot.

In order to facilitate the visualization of treatments in the graphs of the principal components (PCs), these were expressed in a condensed form:

T123

where: 

T: treatment

(1)Corresponds to the ECw levels (ranging from 1 to 2, where 1 = 0.8 dS m^−1^ and 2 = 5.3 dS m^−1^)(2)Corresponds to the H_2_O_2_ concentrations (ranging from 1 to 4, where 1 = 0 μM, 2 = 25 μM, 3 = 50 μM, and 4 = 75 μM)(3)Corresponds to the cotton cultivares (ranging from 1 to 3, where 1= ‘BRS Rubi’, 2 = ‘BRS Topázio’ and 3 = ‘BRS Verde’)

### 3.3. Plant Material

The colored cotton cultivar BRS Rubi was obtained by crossing a material introduced from the USA with dark brown fiber and the CNPA 7H genotype with a white fiber of good quality that is widely adapted to the northeast region. The cultivar BRS Rubi has red brown fiber, a cycle varying between 140 and 150 days, and a mean productivity of 1894 kg ha^−1^ for the Brazilian semi-arid region [43]. BRS Topázio herbaceous cotton has a light brown fiber, a high amount of fibers (43.5%), high uniformity (85.2%), and resistance, conferring excellent characteristics compared to cultivars with white fibers and superior to others with colored fibers [44]. Its mean productivity is 2800 kg ha^−1^, and its cultivation is recommended for the Brazilian northeast, as there is practically no incidence of diseases considering the edaphoclimatic conditions of the region [43]. 

The cultivar BRS Verde is cultivated predominantly in the northeast region due to its wide aptitude for its cultivation in semi-arid regions and the low incidence of diseases in the region. Its cycle is 130–140 days. The thickness of its fiber is close to 30 mm, it has medium fiber characteristics, and it has a length close to that of the white fiber cultivar CNPA 7 H. The fiber strength is approximately 26 g tex^−1^. Its productivity, under rainfed conditions, is approximately 2146 kg ha^−1^, which is slightly below the productivity of white fiber cotton, which reaches 2480 kg ha^−1^ [45]. It is noteworthy that the colored fiber cotton cultivars [46] were classified as moderately sensitive to salt stress during the production formation phase in a previously conducted study.

### 3.4. Experiment Setup and Conduction 

The plants were grown in plastic pots adapted as drainage lysimeters with a capacity of 21 L (35 cm height, 30 cm upper diameter, 20 cm lower diameter), arranged in single rows at a spacing of 0.6 m between rows and 0.3 m between plants in the row. The base of each lysimeter was connected to a drained water collector through a hose with an internal diameter of 3 mm and an external diameter of 5 mm (Figure 3).

At the bottom of each pot, a thin screen was placed, covered by a 500 g layer of crushed stone and 24 kg of soil of a sandy-loam texture. The chemical (Table 4) and physical-hydraulic (Table 5) attributes were analyzed according to Teixeira et al. [47].

Fertilization with nitrogen, phosphorus and potassium was performed as recommended by Novais et al. [48], applying 100 mg of N kg^−1^, 300 mg of P_2_O_5_ kg^−1^, and 150 mg of K_2_O kg^−1^ in the form of urea, monoammonium phosphate and potassium chloride, respectively. Phosphorus was applied as basal fertilization, while N and K were applied as top-dressing, via fertigation, at 30 and 60 days after sowing (DAS). A micronutrient solution was applied monthly at the concentration of 1.0 g L^−1^ of the commercial product Dripsol^®^ micro (SQM VITAS, Candeias, BA, Brazil) containing Mg (1.1%), Zn (4.2%), B (0.85%), Fe (3.4%), Mn (3.2%), Cu (0.5%), and Mo (0.05%), on the adaxial and abaxial sides of the leaves, using a backpack sprayer.

Prior to sowing, the soil moisture was raised to field capacity with locally supplied water (0.28 dS m^−1^), and this water was also used to irrigate the plants until 17 DAS. Sowing was performed by placing five seeds per pot at 1.5 cm depth, distributed equidistantly. At 25 days after germination, the first thinning was carried out, leaving the three most vigorous plants in each lysimeter. At 50 DAS, the second thinning was carried out, leaving only one plant per pot, which was grown until the end of the experiment.

The electrical conductivity levels were determined based on a study conducted by Cavalcante et al. [49], and the solutions were prepared by diluting the salts NaCl, CaCl_2_.2H_2_O, and MgCl_2_.6H_2_O in locally supplied water (0.28 dS m^−1^), with the equivalent ratio of 7:2:1, relative to Na: Ca: Mg, respectively, considering the relationship between ECw and the concentration of salts according to Richards [50], Equation (1).
(1)C=10×ECw
where:

C—concentration of salts to be added (mmol_c_ L^−1^);

ECw—electrical conductivity of water (dS m^−1^).

Hydrogen peroxide (H_2_O_2_) concentrations were established according to a study conducted by Silva et al. [13], prepared by the dilution of H_2_O_2_ in distilled water. The applications, via foliar spraying, were carried out before irrigation with saline water (15 DAS) and were repeated with an interval of 15 days until the opening of the bolls (100 DAS).

Sprays with H_2_O_2_ were carried out using a Jacto XP-12 backpack sprayer, with a pump with working pressure (maximum) of 6 bar, a JD-12 nozzle, and a flow rate of approximately 770 mL/min. Approximately 125 mL of the solution was applied to the plants, initially at three plants per lysimeter, and maintaining this volume after thinning and growth.

Irrigations with saline water started at 18 DAS in order to maintain the soil moisture at a level proportional to the maximum soil holding capacity in all experimental units. The irrigation events were performed manually and daily, applying the volume corresponding to that obtained by the water balance. The volume applied to the plants was determined by Equation (2):(2)$$VI=\frac{\left(Va-Vd\right)}{\left(1-LF\right)}$$
where: 

VI—volume of water to be applied in the next irrigation event (mL);

Va and Vd—volume applied and drained in the previous irrigation event (mL); 

LF—leaching fraction of 0.2 every 15 days.

During the experiment, the cultural practices of manual weeding, surface scarification of the soil, and staking were performed to grow the stem of the plant vertically (Figure 4). For phytosanitary control, insecticides of the chemical group neonicotinoid, fungicides of the chemical group triazole, and acaricide of the chemical group abamectin were applied preventively.

### 3.5. Variables Analyzed

At the end of the crop cycle (130 DAS), according to the methodology of Embrapa cotton, the following variables were quantified: number of bolls (NB), boll weight—g (BW), seed cotton weight—g (SCW), lint cotton weight—g (LCW), percentage of fibers (%F), fiber length—mm (UHM), uniformity—% (UNF), short fiber index (SFI), fiber strength—gf/tex (STR), elongation to rupture—% (ELG), micronaire index (MIC), maturity (MAT), and count strength product index (CSP.

### 3.6. Statistical Analysis

The collected data were subjected to a multivariate analysis, being normalized to a mean of zero (M = 0.0) and a unit variance of (σ^2^ = 1.0). The multivariate composition of the results was evaluated by means of an Exploratory Principal Component Analysis (PCA), condensing the amount of relevant variables contained in the original data set into a smaller number resulting from linear combinations of the original variables generated from the highest eigenvalues (λ > 1.0) in the correlation matrix, explaining a percentage greater than 10% of σ^2^ [51].

The variables of each Principal Component (PC) were subjected to a multivariate analysis of variance (MANOVA) by the Hotelling test at a 0.05 probability level for the factors of H_2_O_2_ concentration, electrical conductivity and cotton cultivares, as well as for their interactions. Only variables with a correlation coefficient greater than 0.5 were maintained in the composition of each principal component [52]. Variables not associated with PCs (r ≤ 0.5) were excluded from the database and subjected to a new analysis. The analyses were processed by Statistica v. 7.0 software [53].

## 4. Conclusions

Irrigation with water of 0.8 dS m^−1^ associated with the foliar application of 75 μM of hydrogen peroxide favored the lint and seed weight, lint weight, strength, micronaire index, and maturity of the fibers in ‘BRS Topázio’, at 130 days after sowing. A water salinity of 0.8 dS m^−1^ associated with foliar applications of 25 and 75 μM of hydrogen peroxide reduced the production and fiber quality in the ‘BRS Verde’ and ‘BRS Rubi’ cultivares, respectively. The ‘BRS Rubi’ cotton cultivar showed higher tolerance to salinity, followed by ‘BRS Topázio’ and ‘BRS Verde’ regarding the yield of seed cotton weight, with a reduction below 20% under water of 5.3 dS m^−1^. The results obtained in this study confirm the hypothesis that hydrogen peroxide, when applied in adequate concentrations, can act as a signaling molecule and reduce the effects of saline stress in colored fiber cotton, with the beneficial effect being mainly dependent on the concentration. However, more studies are needed to understand how hydrogen peroxide acts in salt stress signaling, in addition to validating the results in field research.

## Figures and Tables

**Figure 1 plants-12-02090-f001:**
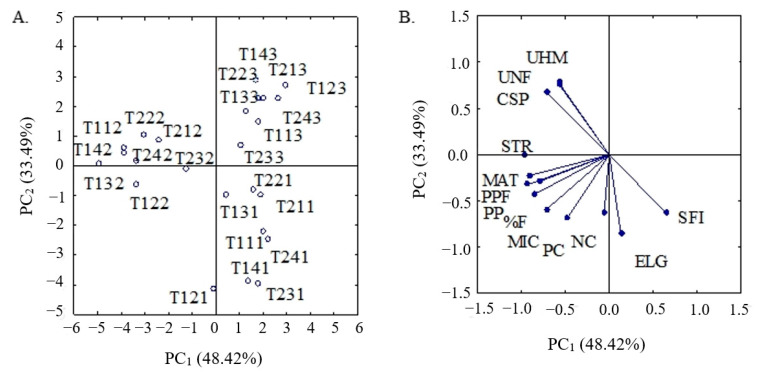
Two-dimensional projection of treatments (**A**) and analyzed variables (**B**) in the two principal components (PC1) and (PC2). T123: T—Treatment; where 1—corresponds to ECw (ranging from 1 to 2: 1 = 0.8 dS m^−1^ and 2 = 5.3 dS m^−1^); 2—corresponds to H_2_O_2_ concentrations (ranging from 1 to 4: where 1= 0 μM, 2 = 25 μM, 3 = 50 μM and 4 = 75 μM); 3—corresponds to cotton cultivares (ranging from 1 to 3: where 1= ‘BRS Rubi’; 2 = ‘BRS Topázio’ and 3 = ‘BRS Verde’). Number of bolls (NB), boll weight (BW), lint and seed weight (LSW), lint weight (LW), percentage of fibers (%F), short fiber index (SFI), strength (gf/tex) (STR), micronaire index (MIC), maturity (MAT), and count strength product index (CSP), fiber length (UHM), percentage of uniformity (UNF), elongation at rupture (ELG).

**Figure 2 plants-12-02090-f002:**
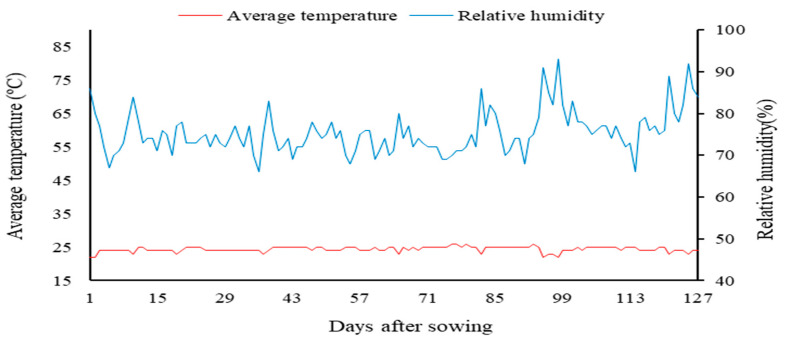
Data of average temperature and relative air humidity observed during the experimental period.

**Figure 3 plants-12-02090-f003:**
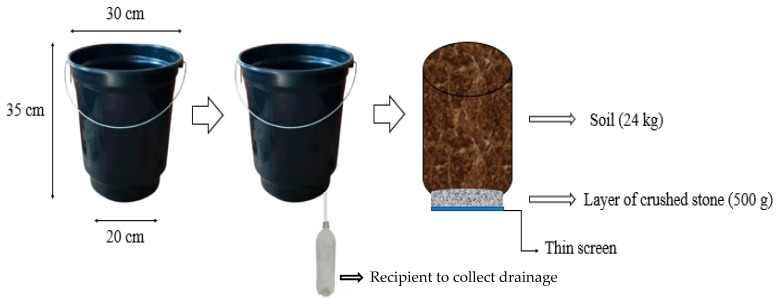
Illustration of filling drainage lysimeters.

**Figure 4 plants-12-02090-f004:**
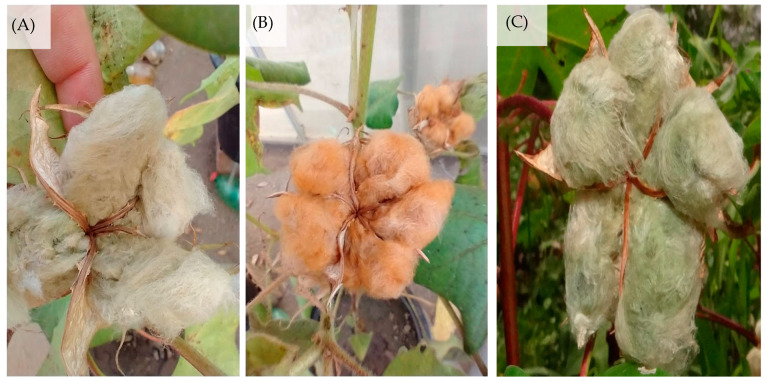
Cultivares of colored fiber cotton (BRS Topázio—(**A**), BRS Rubi—(**B**), and BRS Verde—(**C**)).

**Table 1 plants-12-02090-t001:** Eigenvalues, percentage of total variance explained in the multivariate analysis of variance (MANOVA).

	Principal Components	
	PC1	PC2
Eigenvalues (λ)	6.29	4.35
Percentage of total variance (S^2^%)	48.41	33.49
Hotelling test (T^2^) for electrical conductivity of water (ECw)	0.01	0.01
Hotelling test (T^2^) for H_2_O_2_ concentrations (H_2_O_2_)	0.01	0.01
Hotelling test (T^2^) for cotton cultivares (CC)	0.01	0.01
Hotelling test (T^2^) for interaction (ECw × H_2_O_2_)	0.01	0.01
Hotelling test (T^2^) for interaction (ECw × CC)	0.15	0.14
Hotelling test (T^2^) for interaction (H_2_O_2_ × CC)	0.14	0.11
Hotelling test (T^2^) for interaction (ECw × H_2_O_2_ × CC)	0.54	0.54

Electrical conductivity of water (ECw), concentrations hydrogen peroxide (H_2_O_2_), cotton cultivares (CC).

**Table 2 plants-12-02090-t002:** Correlation coefficients (r) between original variables and the principal components.

PCs	Correlation Coefficients (r)												
	NB	BW	LSW	LW	%F	UHM	UNF	SFI	STR	ELG	MIC	MAT	CSP
PC1	−0.48	−0.05	−0.79	−0.92	−0.84	−0.57	−0.56	0.64	−0.97	0.13	−0.71	−0.89	−0.71
PC2	−0.68	−0.61	−0.29	−0.32	−0.42	0.78	0.75	−0.62	−0.00	−0.84	−0.58	−0.22	0.67

Number of bolls (NB), boll weight (BW), lint and seed weight (LSW), lint weight (LW), percentage of fibers (%F), fiber length (UHM), percentage of uniformity (UNF), short fiber index (SFI), strength (gf/tex) (STR), elongation at rupture (ELG), micronaire index (MIC), maturity (MAT), and count strength product index (CSP).

**Table 3 plants-12-02090-t003:** Mean values of the variables analyzed by treatment.

Treatments	Mean Values												
	NB	BW	LSW	LW	%F	UHM	UNF	SFI	STR	ELG	MIC	MAT	CSP
T111	13	67.63	43.00	15.00	34.88	21.97	81.71	11.76	24.34	5.74	3.05	0.83	2031.57
T121	14	77.03	45.00	18.00	39.29	21.77	82.74	9.31	27.39	8.33	4.27	0.85	2036.96
T131	10	70.00	42.00	15.00	35.71	22.66	83.69	7.46	26.48	6.59	3.68	0.84	2321.98
T141	9	67.56	48.00	17.00	36.36	20.73	80.12	12.95	26.79	7.47	4.11	0.85	1719.56
T211	9	58.47	40.00	14.00	35.00	21.78	82.31	10.31	23.65	5.87	3.61	0.84	2141.50
T221	10	55.89	30.00	11.00	36.67	22.64	83.42	8.12	26.24	6.81	3.93	0.85	2026.73
T231	14	71.13	36.00	13.00	36.11	21.25	81.31	11.81	24.73	7.50	4.23	0.85	1735.48
T241	9	58.62	37.00	14.00	37.50	21.26	81.39	11.95	24.17	7.32	3.89	0.84	1827.09
T112	10	71.46	48.00	21.00	43.75	29.05	86.03	7.05	35.09	5.27	4.22	0.87	3136.91
T122	12	69.34	49.00	23.00	44.90	27.74	85.50	7.67	34.10	6.17	3.98	0.86	3254.80
T132	10	70.00	50.00	22.00	44.64	28.33	86.30	6.42	33.82	6.12	4.08	0.86	3234.27
T142	9	70.00	57.00	24.00	43.10	29.57	86.75	6.37	35.85	5.52	4.65	0.88	3167.87
T212	9	54.59	44.00	20.00	43.24	28.04	84.27	7.28	33.38	5.27	4.14	0.86	3027.94
T222	10	61.46	47.00	20.00	43.75	28.26	86.11	6.65	34.90	5.60	3.94	0.88	3216.45
T232	12	60.31	42.00	17.00	40.48	27.06	85.15	8.18	28.45	5.71	3.98	0.86	2722.56
T242	9	60.12	46.00	20.00	42.55	27.76	84.85	7.00	33.71	5.55	4.75	0.86	2954.89
T113	9	60.90	41.00	12.00	29.27	25.78	84.62	8.27	23.49	5.16	2.61	0.83	2481.62
T123	10	42.76	29.00	9.00	25.64	27.17	84.68	9.48	22.33	4.90	2.59	0.83	2670.60
T133	9	64.03	42.00	12.00	26.47	27.12	85.44	8.22	24.59	5.60	2.65	0.83	2842.11
T143	7	47.98	34.00	10.00	27.78	28.72	86.35	7.59	25.34	6.02	2.84	0.83	2861.56
T213	8	48.08	33.00	9.00	28.57	26.68	83.89	8.98	23.12	5.06	2.70	0.83	2472.97
T223	10	49.72	34.00	9.00	26.47	27.16	84.84	8.22	24.75	5.10	2.89	0.83	2691.24
T233	11	64.63	44.00	12.00	29.03	26.68	84.44	8.45	24.65	5.58	3.08	0.83	2563.64
T243	9	49.63	34.00	10.00	29.41	26.78	85.07	7.56	25.16	5.63	2.86	0.83	2680.36

Number of bolls (NB), boll weight (BW), lint and seed weight (LSW), lint weight (LW), percentage of fibers (%F), short fiber index (SFI), strength (gf/tex) (STR), micronaire index (MIC), maturity (MAT) and count strength product index (CSP), fiber length (UHM), percentage of uniformity (UNF), elongation at rupture (ELG). T123: T—Treatment; 1—corresponds to ECw (ranging from 1 to 2: 1 = 0.8 dS m^−1^ and 2 = 5.3 dS m^−1^); 2—corresponds to H_2_O_2_ concentrations (ranging from 1 to 4: 1= 0 μM, 2 = 25 μM, 3 = 50 μM and 4 = 75 μM); 3—corresponds to cotton cultivares (ranging from 1 to 3: 1 = ‘BRS Rubi’; 2 = ‘BRS Topázio’ and 3 = ‘BRS Verde’).

**Table 4 plants-12-02090-t004:** Chemical attributes of the soil used in the experiment before the application of the treatments.

Chemical Characteristics									
pH (H_2_O) (1:2.5)	OM dag kg^−1^	P (mg kg^−1^)	K^+^	Na^+^	Ca^2+^	Mg^2+^	Al^3+^ + H^+^	ESP (%)	ECse (dS m^−1^)
			(cmol_c_ kg^−1^)						
5.90	1.36	6.80	0.22	0.16	2.60	3.66	1.93	1.87	1.0

OM—Organic Matter: Walkley—lack Wet Digestion; Ca^2+^ and Mg^2+^ extracted with 1 mol L^−1^ KCl at pH 7.0; Na^+^ and K^+^ extracted with 1 mol L^−1^ NH4OAc at pH 7.0; Al^3+^ and H^+^ extracted with 1 mol L^−1^ calcium acetate at pH 7.0; ESP—exchangeable sodium percentage; ECse—electrical conductivity of saturation extract.

**Table 5 plants-12-02090-t005:** Physical-hydraulic attributes of the soil used in the experiment before the application of the treatments.

Physical-Hydraulic Characteristics									
Particle-size fraction (g kg^−1^)			Textural class	Moisture (kPa) dag kg−1		AW	Total porosity %	BD	PD
Sand	Silt	Clay		33.42 *	1519.5 **			(kg dm^−3^)	
732.9	142.1	125.0	SL	11.98	4.32	7.66	47.74	1.39	2.66

AW—available water; BD—bulk density; PD—particle density; SL—sandy loam; *— Field capacity; **—Wilting point.

## Data Availability

Not applicable.

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
