# Peer review of "Production and Fiber Characteristics of Colored Cotton Cultivares under Salt Stress and H2O2"

_plants, 2023, doi:10.3390/plants12112090_

Round 1

Reviewer 1 Report

Although the results of this study was from one cultivation period the aim to evaluate the production and characteristics of fibers of  naturally colored cotton genotypes under irrigation with low- and high-salinity water and leaf application of hydrogen peroxide, is highly interesting. In the introduction a paragraph is needed for the color cotton, culitvation areas, significance of this crop etc in order to explain why the colored cotton are choosen.

Also care have to be taken in below issues

lines 138, 144 replace principal component2 with PC2

line 162: "The micronaire index is a parameter that indicates the behavior and strength of the fibers" is incorrect. the true is that micronaire is an index of thickness and maturity of fiber and also as an index is not measured as "μg pol-1" please check it in lines 163 and 165

line 251: more details about the genetoypes used, are needed

Author Response

Campina Grande, PB

May, 19, 2023

Reference: Plants - 2406336 - Response to Review Report 1

Dear Editor

The authors are very grateful to you and the Reviewers for the positive and constructive comments and suggestions on our manuscript entitled “Production and Fiber Characteristics of Colored Cotton Genotypes Under Salt Stress and H2O2”. The authors would like to inform you that a thorough revision of the manuscript was made, incorporating the suggestions and adopting the text according to the comments. Attached is the revised version of the manuscript. All changes in the text are highlighted in red color.

The authors remain at your disposal for any further information and explanation.

The responses/clarifications to the issues raised by the Reviewer 1/Editor are presented below:

REVIEWER 1

  1. Although the results of this study was from one cultivation period the aim to evaluate the production and characteristics of fibers of naturally colored cotton genotypes under irrigation with low- and high-salinity water and leaf application of hydrogen peroxide, is highly interesting. In the introduction a paragraph is needed for the color cotton, culitvation areas, significance of this crop etc in order to explain why the colored cotton are choosen.

Response: Dear reviewer, we thank you for the excellent suggestion, we inform you that the introduction was reformulated in the revised version of the manuscript, taking into account the suggestions of the reviewers (Lines 39 – 51).

  1. Lines 138, 144 replace principal component2 with PC2 (as linhas 138, 144 substituem o componente principal2 por PC2).

Response: The terms have been replaced in the revised version of the manuscript, as can be seen in lines 151 and 157.

  1. Line 162: "The micronaire index is a parameter that indicates the behavior and strength of the fibers" is incorrect. The true is that micronaire is an index of thickness and maturity of fiber and also as an index is not measured as "μg pol-1" please check it in lines 163 and 165.

Response: The text was reformulated as suggested by the reviewer and the unit μg in-1 was removed, as can be seen between lines 173 and 179, of the revised version of the manuscript.

  1. Line 251: more details about the genetoypes used, are needed

Response: The suggestion was accepted, in the revised version of the manuscript. A topic 3.3 was added with the details of the genotypes used in the research (Lines 285-304).

Yours sincerely,

Geovani Soares de Lima

Reviewer 2 Report

The manuscript of Luana Lucas de Sá Almeida Veloso , Carlos Alberto Vieira de Azevedo, Reginaldo Gomes Nobre, Geovani Soares de Lima, José Renato Cortez Bezerra , André Alisson Rodrigues da Silva, Reynaldo Teodoro de Fátima , Hans Raj Gheyi, Lauri ane Almeida dos Anjos Soares, Pedro Dantas Fernandes , Vera Lúcia Antunes de lima and Lúcia Helena Garólafo Chaves “Production and Fiber Characteristics of Colored Cotton Genotypes Under Salt Stress and H2O2”,dedicated to the actual subject : salt stress management for one of the most important agricultural crops, Gossypium hirsutum L.

Authors obtained Interesting results that could be interesting for a wide range of scientific community working in the area of plant’s salt tolerance problem.

Comments:

1. The abstract does not reflect the essence of the work carried out by the authors, it needs to be

rewrite

2 It’s not clear the choice of precisely colored cotton varieties as objects for ongoing research. In the introduction, the authors mention white cotton, but they take in the experiment colored cotton varieties

3. It is not clear why exactly the varieties BRS Rubi’, ‘BRS Topázio’ and ‘BRS Verde’ were chosen for the study. Аre these highly productive regional varieties?

4 Why the authors didn’t include in experiment a standard variety of colored cotton (highly productive variety adapted to local conditions) to compare the indicators obtained for the varieties of colored cotton taken in the study with indicators of standard variety

5. The authors conducted the experiment in a greenhouse then under what conditions? In the section “Materials and Methods” authors given the climatic conditions. How outside climatic conditions effect the experimental data obtained in the greenhouse?

Recommendations

1. Divide the table 1 into several tables,

a large number of heterogeneous results given in one table hinders understanding

2. the same goes for the second table

Author Response

Campina Grande, PB

May, 19, 2023

Reference: Plants - 2406336 - Response to Review Report 2

Dear Editor

The authors are very grateful to you and the Reviewers for the positive and constructive comments and suggestions on our manuscript entitled “Production and Fiber Characteristics of Colored Cotton Genotypes Under Salt Stress and H2O2”. The authors would like to inform you that a thorough revision of the manuscript was made, incorporating the suggestions and adopting the text according to the comments. Attached is the revised version of the manuscript. All changes in the text are highlighted in red color.

The authors remain at your disposal for any further information and explanation.

The responses/clarifications to the issues raised by the Reviewer 2/Editor are presented below:

REVIEWER 2

  1. The abstract does not reflect the essence of the work carried out by the authors, it needs to be rewrite.

Response: Dear reviewer, the abstract was reformulated in the revised version of the manuscript, inserting the context of the problem addressed in the study and the possible solution to alleviate the damage caused by salt stress. However, it is worth mentioning that, according to norms of Plants, the abstract should not contain more than 200 words, for this reason it was not possible to insert more information.

  1. It’s not clear the choice of precisely colored cotton varieties as objects for ongoing research. In the introduction, the authors mention white cotton, but they take in the experiment colored cotton varieties

Response:  Dear reviewer, in the revised version of the manuscript, the introduction has been reformulated, inserting information about colored fiber cotton. In addition, in the material and methods section a topic 3.3 has been included with details of the colored cotton varieties under study.

  1. It is not clear why exactly the varieties BRS Rubi’, ‘BRS Topázio’ and ‘BRS Verde’ were chosen for the study. Аre these highly productive regional varieties?

Response: The justification for choosing the varieties was given by the wide aptitude for their cultivation in semi-arid regions and the low incidence of diseases, in topic 3.3 of the revised version of the manuscript, some details of the choice of varieties has been  inserted as well.

  1. Why the authors didn’t include in experiment a standard variety of colored cotton (highly productive variety adapted to local conditions) to compare the indicators obtained for the varieties of colored cotton taken in the study with indicators of standard variety.

Response:  Dear reviewer, as mentioned in topic 3.3 of the revised version of the manuscript, the choice of colored cotton varieties was based on productivity characteristics, adaptation to local conditions and disease incidence.

  1. The authors conducted the experiment in a greenhouse then under what conditions? In the section “Materials and Methods” authors given the climatic conditions. How outside climatic conditions effect the experimental data obtained in the greenhouse?

Response:  Dear reviewer, it is worth highlighting that cultivation in a protected environment allows a number of advantages for crops, for example: less leaching of nutrients from the soil, more efficient control of pests and diseases, protection against weather conditions such as rain, wind, hail, cold and high rates of radiation and precocity, consequently generating improved product quality and increased productivity. However, under conditions of protected environment, the temperature indexes are higher and the relative humidity is lower in relation to the external environment. Having obtained promising results under controled conditions, future studies will be conducted under field conditions.

  1. Divide the table 1 into several tables, a large number of heterogeneous results given in one table hinders understanding.

Response: In the revised version of the manuscript, Table 1 is divided into three tables, as suggested by the reviewer.

  1. The same goes for the second table.

Response: The suggestion was accepted in the revised version of the manuscript.

Yours sincerely,

Geovani Soares de Lima
